# Recent Advances in Bacterial Amelioration of Plant Drought and Salt Stress

**DOI:** 10.3390/biology11030437

**Published:** 2022-03-12

**Authors:** Elisa Gamalero, Bernard R. Glick

**Affiliations:** 1Dipartimento di Scienze e Innovazione Tecnologica, Università del Piemonte Orientale, Viale T. Michel 11, 15121 Alessandria, Italy; 2Department of Biology, University of Waterloo, Waterloo, ON N2L 3G1, Canada; glick@uwaterloo.ca

**Keywords:** sustainable agriculture, plant growth-promoting bacteria (PGPB), salt stress, drought stress

## Abstract

**Simple Summary:**

Salt and drought stress cause enormous crop losses worldwide. Several different approaches may be taken to address this problem, including increased use of irrigation, use of both traditional breeding and genetic engineering to develop salt-tolerant and drought-resistant crop plants, and the directed use of naturally occurring plant growth-promoting bacteria. Here, the mechanisms used by these plant growth-promoting bacteria are summarized and discussed. Moreover, recently reported studies of the effects that these organisms have on the growth of plants in the laboratory, the greenhouse, and the field under high salt and/or drought conditions is discussed in some detail. It is hoped that by understanding the mechanisms that these naturally occurring plant growth-promoting bacteria utilize to overcome damaging environmental stresses, it may be possible to employ these organisms to increase future agricultural productivity.

**Abstract:**

The recent literature indicates that plant growth-promoting bacteria (PGPB) employ a range of mechanisms to augment a plant’s ability to ameliorate salt and drought stress. These mechanisms include synthesis of auxins, especially indoleacetic acid, which directly promotes plant growth; synthesis of antioxidant enzymes such as catalase, superoxide dismutase and peroxidase, which prevents the deleterious effects of reactive oxygen species; synthesis of small molecule osmolytes, e.g., trehalose and proline, which structures the water content within plant and bacterial cells and reduces plant turgor pressure; nitrogen fixation, which directly improves plant growth; synthesis of exopolysaccharides, which protects plant cells from water loss and stabilizes soil aggregates; synthesis of antibiotics, which protects stress-debilitated plants from soil pathogens; and synthesis of the enzyme 1-aminocyclopropane-1-carboxylate (ACC) deaminase, which lowers the level of ACC and ethylene in plants, thereby decreasing stress-induced plant senescence. Many of the reports of overcoming these plant stresses indicate that the most successful PGPB possess several of these mechanisms; however, the involvement of any particular mechanism in plant protection is nearly always inferred and not proven.

## 1. Drought and Salt Stress

Plants that are grown in the field under natural conditions are typically subjected to a wide range of both biotic and abiotic stresses, any one of which may significantly inhibit plant growth and development [1]. Biotic stress factors that are inhibitory to plant growth and development include viruses, nematodes, insects, phytopathogenic bacteria, and phytopathogenic fungi. In addition, abiotic stress factors that are detrimental to plants include extremes of temperature, excessively high or low light conditions, flooding, drought, high salt, toxic metals, organic contaminants, and excessive radiation. This review focuses on the effects and remediation of drought and salt stress, two of the most problematic abiotic stresses as far as plant growth is concerned.

Soil salinity and drought are enormous global problems for the growth of agricultural plants. Worldwide drought has increased dramatically in recent years because of the increasing impacts of climate change (https://climate.nasa.gov/news/3117/drought-makes-its-home-on-the-range/, accessed on 5 January 2022; https://www.c2es.org/content/drought-and-climate-change/, accessed on 5 January 2022). Soil salinity is not only affected by climate change but is especially problematic for crops that require irrigation. Because of the lack of sufficient water (i.e., drought) in many locations, some farmlands are under-irrigated, causing salt (from the irrigation water) to accumulate in the soil. In this case, the salt remains in the soil when the irrigation water is either utilized by plants or is lost to evaporation. Moreover, the majority of the world’s salt-inhibited soils are found in arid or semi-arid climates (https://www.fao.org/global-soil-partnership/resources/highlights/detail/en/c/1412475/, accessed on 5 January 2022). Compounding the negative effects of drought, the high salt levels that result from irrigation are inhibitory to the growth of a large number of plants. At present, it is estimated that >900 million hectares or ~20% of the world’s cultivated land mass is negatively impacted by high levels of salt. Moreover, a significant fraction of the global land mass used for the growth of irrigated crops is at least somewhat adversely affected by high levels of salinity.

The initial responses of most plants to drought and salinity are similar; both responses are largely due to water stress within the plant [2,3]. When plants are first exposed to high levels of salt, a decrease in their growth rate occurs. This is often followed by a slow recovery to a lower growth rate. Subsequently, following the continued uptake of salt by plants, sodium ions are translocated through the xylem to the plant shoots, with leaves and shoots eventually accumulating high levels of sodium ions. The toxicity of accumulating sodium ions in plants is generally considered to be a function of the excessive amounts of sodium ions that compete with potassium ions for the binding sites that are necessary for plant cellular functioning. This abiotic stress results in the generation of a number of reactive oxygen species within the plant [4], which in turn can cause electrolyte leakage from plant cells, plant cell membrane lipid peroxidation, an increase in photorespiration, a decrease in transpiration, eventual pollen sterility, and a decrease in the plant’s rate of photosynthesis, all of which negatively affect the plant yield and quality.

Plants are able to use a range of mechanisms to deal with salt stress [2,5,6,7,8,9,10,11,12,13,14], including (i) selective accumulation (in vacuoles) or exclusion of sodium ions, (ii) modulation of the root uptake of sodium ions and the subsequent transport of these ions into leaves, (iii) compartmentalization of sodium ions at the plant cellular level, (iv) synthesis of a variety of compatible solutes such as trehalose and proline, (v) modification of plant cell membranes, (vi) synthesis of various antioxidative enzymes, including superoxide dismutase and peroxidase, and (vii) modulation of some plant hormone levels, including auxin, cytokinin, and ethylene. Moreover, the salt tolerance of any particular plant species is often a function of the specific salt tolerance of the cultivar of the plant being examined, the growth phase of the plant, the soil composition, the plant’s health, the presence and nature of plant pathogenic organisms, the presence of specific rhizospheric or endophytic plant growth-promoting bacteria (PGPB), and the presence of mycorrhizae [1].

In considering the different PGPB mechanisms that can be employed (elaborated in the following section), it is necessary to bear in mind that many of these activities and their regulation are interconnected. Moreover, different PGPB contain different sets of genes that enable them to provide a range of varied protective responses against the inhibitory effects of a range of abiotic (and biotic) stresses, including drought and salinity. In addition to the more direct effects of PGPB on plants, many PGPB can modify plant gene expression, thereby increasing the plant’s synthesis of stress protective agents. Thus, for example, some PGPB may promote an increase in the plant’s production of “water-structuring” metabolites (osmolytes), such as betaine, proline, and trehalose, and the synthesis of reactive oxygen detoxifying enzymes, such as superoxide dismutase and catalase [15].

In assessing the scientific literature, it should be pointed out that the ability of PGPB to overcome the effects of salt stress on plant growth has been studied to a much greater extent than the ability of PGPB to ameliorate drought stress. This somewhat disparate focus may reflect the fact that salt stress is technically easier to impose and study in a laboratory setting and is of more universal interest than drought stress. This notwithstanding, numerous studies have reported success in using PGPB to ameliorate some of the deleterious effects of both salt and drought stress.

Interestingly, drought and/or salt stress often increases the sensitivity of many plants to various phytopathogens, often by decreasing the plant’s ability to effectively mount an attack against the pathogen. However, it has been observed that PGPB isolated from salt- or drought-stressed soils, in addition to the previously mentioned ability to protect plants against these abiotic stresses, can frequently effectively protect plants against the damage caused by many different fungal phytopathogens [16,17,18].

In the examples given below, conferring plants with drought and/or salt tolerance is often attributed to one or more well-known PGPB mechanisms; this is despite the fact that drought and/or salt tolerance in nature is probably attributable to multiple mechanisms. In addition, the scientific literature also contains numerous studies in which the mechanism used by a PGPB strain to confer drought and/or salt tolerance to a plant is not known. While these strains may be useful in some limited applications, since other researchers do not know what mechanisms they are using, the isolated strains are only useful to those who have isolated and identified them, since they do not provide any guidelines for selecting other (possibly similar or more efficacious) PGPB that are better at conferring salt and/or drought tolerance to treated plants.

## 2. Plant Growth-Promoting Bacteria

A very large number of bacteria are typically found in soil (~10^8^ to 10^9^ bacterial cells per gram of soil), although drought- or salt-stressed soils often contain as little as 10^6^ bacterial cells per gram of soil [19]. Moreover, the bacteria are not evenly distributed throughout the soil. That is, much greater numbers of bacteria are usually found around the roots of plants than in the bulk soil. This is a consequence of the fact that plant roots generally exude a large amount sugars, amino acids, and small organic acids, and different types of these small molecules can serve as a nutrient source for many different soil bacteria [20]. The bacteria in the soil are made up of a range of different genera and species, some of which can promote plant growth (i.e., PGPB), some of which can inhibit plant growth (i.e., phytopathogens), and some of which have no discernible effect on plant growth.

PGPB stimulation of plant growth by may occur by a variety of either direct or indirect mechanisms. The direct promotion of plant growth by PGPB may occur by the bacterium, providing a plant with one or more phytohormones, such as auxin, cytokinin, or gibberellin, or by improving the ability of a plant to acquire soil nutrients by providing plants with sequestered iron, solubilized phosphate, zinc and potassium, and fixed nitrogen. In addition, PGPB that express the enzyme 1-aminocyclopropane-1-carboxylate (ACC) deaminase can lower plant ACC and ethylene levels and hence significantly increase a plant’s tolerance to a wide range of environmental ethylene-generating stresses [11,21,22,23,24]. Importantly, PGPB may possess one or more of these activities, with most PGPB possessing a few of these activities.

The indirect promotion of plant growth by PGPB happens when the PGPB acts as a biocontrol agent and can prevent or thwart phytopathogens (usually either fungi or bacteria) from inhibiting plant growth. This may occur in several different ways, including the synthesis of pathogen-inhibiting antibiotics and/or hydrogen cyanide by the PGPB; the synthesis of any one of several different pathogenic fungal cell wall lytic enzymes; the synthesis of siderophores that bind extremely tightly to soil Fe^+++^, resulting in the pathogen (that has a much lower affinity for iron) being deprived of necessary iron; the PGPB outcompeting the phytopathogen for sites on or near the plant root; the synthesis of plant hormones by the PGPB, thereby promoting plant growth and strengthening the plant’s ability to fight off the attacking phytopathogen; the PGPB decreasing the amount of pathogen-induced stress ethylene within a plant; the synthesis of volatile organic compounds (VOC) that are toxic to a variety of pathogenic organisms; and the PGPB inducing the plant’s own defenses against phytopathogens (i.e., Induced Systemic Resistance) [1].

Several of the direct mechanisms mentioned above have previously been shown to protect plants against both drought and salt stress [10,13,25,26,27,28]. For example, cytokinin, auxin, and ethylene levels have all been directly implicated in playing a role in how plants respond to drought and salt stress [1,2,10,22,29]. In addition, indirect mechanisms of plant growth promotion have been shown to protect a salt- or drought-debilitated plant against the phytopathogens present during periods of salt or drought stress [18].

PGPB and the mechanisms that they employ have been extensively studied over the past 20–25 years, and they have been shown to be effective at facilitating plant growth and development under a wide range of potentially inhibitory conditions, with many different species of plants, and under laboratory, greenhouse, and field conditions. Moreover, several of the isolated and characterized PGPB strains have been commercialized in several different countries [1,30]. Although PGPB presently make up only a very small portion of the global fertilizer market, with each succeeding year, more of these bacteria are being used in sustainable agricultural practice. It is therefore expected that PGPB will eventually (hopefully in the next 5–15 years) be utilized on a global scale and replace many of the environmentally deleterious chemicals that are currently used in agriculture.

### 2.1. How PGPB Mechanisms Deal with Plant Drought and Salt Stress

#### 2.1.1. Mechanisms: ACC Deaminase, IAA, Cytokinin, and Metabolites Such as Proline and Trehalose

PGPB exhibit a wide range of mechanisms through which they can improve the tolerance of plants to salt and drought stress. After a careful analysis of the existing literature, some of the most common traits involved are (i) the synthesis of the enzyme ACC deaminase, lowering the amount of stress ethylene produced by plants; (ii) the release of auxins and especially IAA, modulating the regulatory responses of plants that have been exposed to various environmental stresses; (iii) the synthesis of metabolites such as proline and trehalose, well known to behave as protectants against various stresses (particularly drought and salt stress; Figure 1). The protective effect of these bacterial mechanisms against abiotic stresses has been extensively tested and validated for a large number of plant and bacterial species. In the following section, a description of the functioning of these mechanisms is presented.

#### 2.1.2. Details of How Each Mechanism Functions

The plant hormone ethylene is involved in several plant biological activities, including germination of seeds, tissue differentiation, the regulation of root branching and root elongation, flowering, fruit ripening, and leaf abscission. However, when plants are subjected to biotic or abiotic environmental stresses, the synthesis of a high amount of ethylene occurs. This synthesis starts from S-Adenosyl-Met; this molecule is converted into ACC by ACC synthase, and the ACC is subsequently converted into ethylene by the enzyme ACC oxidase [31]. ACC deaminase produced by PGPB can lower ACC and ethylene levels in a plant via the degradation of ACC to ammonia and alpha-keto-butyrate. As a consequence, the ethylene level inside a plant will not reach concentrations that are inhibitory to plant growth [32,33]. There is a body of literature demonstrating the efficacy of PGPB in producing ACC deaminase [34], supporting plant growth in the presence of salt or/and drought stresses [1,2] (see Table 1 and Table 2). The clear involvement of ACC deaminase in facilitating plant growth under water deprivation or the excess of salt is typically demonstrated with the use of mutants lacking this enzymatic activity [25,34]. Moreover, ACC deaminase is often associated with the expression of other PGPB plant-beneficial traits, such as IAA synthesis [33], and the production of osmoprotectant molecules [35]. Auxins are phytohormones whose effect is dose-dependent, driving different stages of plant development, such as germination of seeds, xylematic vessel formation, the elongation and proliferation of plant cells, root branching (including the emergence of both lateral and adventitious roots), plant responses to light and gravity, photosynthesis, florescence and fructification, and tolerance to stressful conditions [36]. Soil bacteria able to synthesize IAA are quite common and belong to a wide variety of taxonomic groups. Bacterial biosyntheis of IAA may occur through one or more of seven different biosynthetic pathways that have been identified, with five of these pathways relying on tryptophan as a precursor of IAA [37,38]. Usually, since amino acid synthesis is costly from an energetic point of view, the tryptophan concentration inside a bacterial cell is low and is strictly regulated. For this reason, to synthesize IAA, bacterial cells primarily use the tryptophan released through exudation by the plant roots. IAA produced by bacteria may be taken up by plant roots, augmenting the endogenous plant pool. The increased IAA generally has the effect of stimulating plant growth (or suppressing plant growth if the IAA level becomes too high).

IAA-producing bacteria mediate drought and salt stress tolerance by improving root structure architecture, increasing the permeability of water into the cell and the water uptake in leaves, regulating metabolic homeostasis by increasing osmotic content, ROS detoxification, and inducing the transcription of a plethora of stress-related genes combined with the induction of specific protein synthesis [39,40].

Following several previous experiments using transgenic plants, a strain of the Gram-negative bacterium *Ensifer meliloti* was engineered with an *ipt* gene from a strain of *Agrobacterium* under the control of the *E. coli trp* promoter, which causes the overproduction of the phytohormone cytokinin [41]. The genetically engineered bacteria were then assessed to determine whether they could protect alfalfa plants against the growth inhibition and senescence that is a consequence of drought stress. Alfalfa plants were first inoculated with engineered *E. meliloti* and then subjected to several weeks of drought stress. Following this stress, the plants inoculated with the transformed PGPB strain were significantly larger than the plants inoculated with the non-transformed strain. When the plants were later rewatered, those that had been inoculated with the engineered PGPB strain grew to a level similar to plants that had not been subjected to drought stress, suggesting that the high levels of bacterial cytokinin produced by the engineered bacterium improved the ability of alfalfa to withstand drought stress. Unfortunately, the use of genetically engineered bacterial strains in the field is not permitted in most countries; however, it may be possible to alter rhizobial strains to produce high cytokinin levels by either conventional mutagenesis or CRISPR modification of the bacterial genome (which is not considered to be genetic engineering in many countries).

Trehalose (α-D-glucopyranosyl-(1→1)-α-D-glucopyranoside) is a non-reducing disaccharide that consists of two glucose molecules linked by a α,α-1,1-glycosidic bond; it is found in archaea, bacteria, fungi, plants, and in many invertebrates. In bacteria, trehalose synthesis is realized through five different pathways, with only one consisting of a single reaction, e.g., in *Pseudomonas putida* UW4 and other bacterial species, and consisting of trehalose synthesis from maltose mediated by a transglucosylation reaction [42]. Trehalose in plant and bacterial cells behave as a xeroprotectant, through three hypothesized mechanisms (water replacement, glass formation, and stability theory), enabling them to deal with harsh abiotic stress such as desiccation and or high levels of salinity [43,44,45,46]. The environmental pressure induced by salinity or drought selects and favors tolerant plants and microorganisms. As a consequence, PGPB living on the root and able to tolerate arid or saline conditions can limit the damage caused by drought and/or salinity to the plants [47]. One of the mechanisms responsible for the induced plant tolerance to desiccation is the bacterial modulation of the amount of trehalose in plants. Trehalose behaves as a signal of drying damage due to its ability to trigger the plant defense system to limit the damages caused by drought. This was demonstrated by Vilchez et al. [48], who identified the *ots*AB genes (which codes for alpha, alpha-trehalose-phosphate synthase, and trehalose-6-phosphate phosphatase) in the drought-tolerant *Microbacterium* sp. strain 3J1 and inserted them into the drought-sensitive *P. putida* KT2440 strain. The transformed strain *P. putida* KT2440 (pUCP22:*ots*AB) overproduced trehalose under water stress conditions and efficiently supported the growth of pepper plants cultivated under water deficit. Moreover, several examples in the literature, reviewed by Sharma et al. [49], demonstrate that trehalose is a key component of the functioning of the legume-rhizobia-mycorrhizal fungi tripartite symbiosis under drought conditions.

In addition to increasing bacterial and plant tolerance to environmental stresses, trehalose improves the survival of PGPB or other bacterial biocontrol agents during long-term storage of commercial formulations and increases root competence [50]. Finally, trehalose has recently been classified by the US Food and Drug Administration as Generally Regarded As Safe (GRAS) [51], thus becoming useful as a food formulant.

Among the compatible solutes produced by the plant following water deprivation, proline reduces the cell water potential and helps to maintain the turgor pressure, thus ensuring the plant’s full range of plant metabolic activity, development, and yield rate. The amount of proline synthesized by different species of plants increases from 20% to 80% of the total number of free amino acids under optimal conditions and osmotic and salt stress, respectively [52]. Fortunately, PGPB can modulate proline expression in plants. For example, *Pseudomonas putida* strain GAP-P45 can promote the growth of *Arabidopsis thaliana* plants that have been exposed to drought, upregulating the expression of genes that are involved in proline production, including OAT (ornithine-Δ-aminotransferase), P5CS1 (Δ1-pyrroline-5-carboxylate synthetase 1), P5CR (Δ1-pyrroline5-carboxylate reductase), and in proline degradation, such as PHD1 (proline dehydrogenase 1) and P5CDH (Δ1–pyrroline 5-carboxylate dehydrogenase). These results, obtained by real-time PCR, are consistent with the improvement of the growth parameters (plant biomass, water content, chlorophyll concentration) observed in plants cultivated under water deficit and inoculated with this bacterial strain [53]. Similarly, salt tolerance in wheat induced by *Bacillus* sp. strain wp-6 was determined to be related to an increased content of proline, soluble sugar, and soluble protein of 7.48%, 12.34%, and 4.12%, respectively [54].

**Table 1 biology-11-00437-t001:** Recent studies indicating how PGPB help to ameliorate drought stress.

Plant	Bacteria	Comments	Reference
*Arabidopsis thaliana* (thale cress)	*Flavobacterium crocinum* HYN0056T	Inoculation with HYN0056T enhanced tolerance against drought and salt stress, possibly via induction of stomatal closure. Treatment with strain HYN0056T followed by drought or salt stress caused the upregulation of several drought- and salt-inducible *Arabidopsis* genes.	[55]
*Arabidopsis thaliana* (thale cress)	*Kosakonia cowanii* GG1 isolated as an endophyte of seeds of the xerophytic invasive plant *Lactuca serriola*	Inoculation of *A. thaliana* with *K. cowanii* GG1 stimulated plant growth under drought conditions. The bacterial strain reduced soil water loss, indicating that the synthesis of exopolysaccharides contributes to maintaining the soil water content.	[56]
*C. arietinum* L. (chickpea)	*Mesorhizobium ciceri* Ca181	Mutants of *M. ciceri* defective in phosphorous solubilization and drought stress tolerance were selected. The results indicated that the *otsA* (Trehalose-6-phosphate synthase), *Auc* (Acetoin utilisation protein), and *Usp* (Universal stress protein) genes contributed to the mechanism of drought stress tolerance.	[57]
*Eleusine coracana* (L.) Gaertn. (finger millet)	*Variovorax paradoxus* RAA3 and ACC deaminase-producing bacteria (*Ochrobactrum anthropi* DPC9, *Pseudomonas palleroniana* DPB13 and DPB16, and *Pseudomonas fluorescens* DPB15)	Inoculation of plants with *V. paradoxus* RAA3 and the consortium of *O. anthropi* DPC9, *P. palleroniana* DPB13 and DPB16, and *P. fluorescens* DPB15 increased plant growth and nutrient levels in leaves. High amounts of ROS-scavenging enzymes, including superoxide dismutase, guaiacol peroxidase, catalase, and ascorbate peroxidase, as well as the cellular osmolytes proline and phenol, leaf chlorophyll, and a reduced level of hydrogen peroxide and malondialdehyde, were observed after inoculation with RAA3 and the consortium of the other four bacterial strains compared to untreated plants.	[58]
*Eucalyptus* *grandis* (rose gum)	*Pseudomonas* sp. M25 and N33	Plants inoculated with strain M25 exposed to gradual water deficit showed a significant increase in plant water content and cell wall elasticity. Rapid water deficit conditions caused partial defoliation in the absence of added bacteria. Both PGPB strains stimulated the formation of new leaves; inoculation with strain M25 reduced the transpiration rate; and co-inoculation with both strains increased both growth and photosynthetic activity.	[59]
*Glycine max* L. Merrill (soybean)	*Bradyrhizobium japonicum* and *Azospirillum brasilense*	The inoculation of soybean plants with *B*. *japonicum* and/or *A. brasilense* and then subjected to drought stress yielded increased leaf membrane stability. Co-inoculation with these strains followed by drought stress improved nodulation. Treatment with one or both of these strains reduced the pod abortion rate under moderate drought stress but not severe drought stress.	[60]
*Juglans regia* L. (walnut.)	AM fungi (*Funnelliformis mosseae**,* *Claroideoglomus etunicatum*), and *Azotobacter chroococcum, Azospirillium lipoferum*	Drought stress caused a reduction in plant growth and leaf nutrient content, while increased proline, soluble sugar, starch peroxidase enzyme activity, and phenolic content was seen in leaves. Inoculation with consortia alleviated the negative effects of drought stress on seedlings by increasing the phenol, proline, peroxidase activity, soluble sugar, and starch content. *C. etunicatum* was the most effective AM fungi.	[61]
*Solanum lycopersicum* Mill *cv*. F144 (tomato) and *Capsicum annuum* L. *cv.* Maor (pepper)	*Achromobacter piechaudii* ARV8 (isolated from the Arava region of the Negev desert, Israel)	Tomato and pepper seedlings exposed to transient water stress and inoculated with strain A. *piechaudii* ARV8 showed increased biomass. Moreover, ARV8 lowered ethylene synthesis in tomato seedlings during drought stress, favoring the recovery of ARV8-treated plants when watering was resumed.	[62]
*Solanum lycopersicum* L. (tomato)	Various rhizobacteria (12 *Bacillus* spp. strains, 6 *Pseudomonas*, 4 *Brevibacillus* and 1 *Paenibacillus* strain isolated from *Cistanthe longiscapa* (a native flowering desert plant from the Atacama desert, Chile)	The *Bacillus* strains were used to formulate three consortia and to inoculate tomato seeds that were subsequently exposed to different degrees of water limitation. Inoculated seedlings showed higher biomass and recovery rates compared to uninoculated ones.	[63]
*Solanum tuberosum* (cultivars Swift and Nevsky) (potato)	*Achromobacter xylosoxidans* Cm4, *Pseudomonas oryzihabitans* Ep4, and *Variovorax paradoxus* 5C-2	PGPB inoculation increased tuber yield in field experiments in plants cultivated under water-limited conditions. However, the leaf water concentration both in inoculated and uninoculated plants was similar, suggesting that other mechanisms (such as the modulation of phytohormone levels) might be responsible for plant growth promotion.	[64]
*Triticum aestivum* L. (wheat)	*Azospirillum brasilense* and *Herbaspirillum seropedicae*	Inoculation with the bacterial strains induced drought resistance in the wheat cultivar CD-120. The grain index was improved with *H. seropedicae* under water stress conditions.	[65]
*Triticum aestivum* L. (wheat)	*Variovorax paradoxus* RAA3; *Pseudomonas* spp. DPC12, DPB13, DPB15, and DPB16; *Achromobacter* spp. PSA7 and PSB8; and *Ochrobactrum anthropi* DPC9	In drought conditions, inoculation with strain RAA3 and a consortium of DPC9 + DPB13 + DPB15 + DPB16 improved wheat plant growth and foliar nutrient levels and positively modulated antioxidant properties compared to uninoculated plants.	[66]
*Triticum aestivum* L. (wheat)	*Azospirillum brasilense* NO40 and *Stenotrophomonas maltophilia* B11	Seedling inoculation with the two bacterial strains overcame the negative effects of drought stress, including changes to the relative water content of roots, shoots, and leaves; the area of leaves; the contents of chlorophyll a and b and ascorbic acid; and the protein patterns of root extracts. Bacterial inoculation reduced the drought-induced negative changes (i.e., the leakage of electrolytes and accumulation of malondialdehyde and hydrogen peroxide, the production of proline, and the activities of catalase and peroxidase compared to their uninoculated counterparts).	[67]
*Triticum aestivum* (wheat)	*Curtobacterium flaccumfaciens Cf* D3-2 and *Arthrobacter* sp. *Ar* sp. D4-1	The two bacterial strains, when used separately to inoculate wheat plants, showed the ability to promote growth under drought conditions.	[68]
*Vigna mungo* L. (black gam) and *Pisum sativum* L. (pea)	*Ochrobactrum pseudogrignonense* RJ12, *Pseudomonas* sp. RJ15, *Bacillus subtilis* RJ46, alone and in consortia	Consortium treatment increased seed germination, root and shoot length, and plant biomass. Under drought conditions, treated plants exhibited elevated ROS and cellular osmolyte synthesis, higher leaf chlorophyll content, and increased relative water content compared to uninoculated plants. Bacterial inoculation reduced ACC accumulation in plants and down-regulated ACC-oxidase gene expression.	[69]
*Vigna unguiculata* (cowpea)	*Bacillus aryabhattai* strain MoB09 (able to degrade the herbicide paraquat)	*B. aryabhattai* MoB09 promoted the growth of cowpea plants following drought stress.	[70]
*Zea mays* L. (maize)	12 drought-tolerant bacterial strains producing ACC deaminase and/or exopolysaccharides	Strains that synthesize both ACC deaminase and exopolysaccharides induced increased photosynthesis rate, stomatal conductance, vapor pressure, water-use efficiency, and transpiration rate. The strain *B. velezensis* D3 was the best PGPB.	[71]
*Zea mays* L. (maize)	Commercial biostimulant BACSTIMR 100 composed of a consortium of two *Bacillus licheniformis* strains, two *Brevibacillus laterosporus* strains, and one *Bacillus amyloliquefaciens* strain	Plant inoculation conferred increaseed drought resistance in maize by altering several plant metabolic pathways, including pathways encoding redox homeostasis and strengthening of the plant cell wall, osmoregulation, energy production, and membrane remodelling.	[72]

**Table 2 biology-11-00437-t002:** Recent studies indicating how PGPB help to ameliorate salt stress.

Plant	Bacteria	Comments	Reference
*Arabidopsis thaliana* (thale cress)	*Paraburkholderia phytofirmans* PsJN	*A. thaliana* plants inoculated with strain PsJN showed higher survival rate when exposed to long-term salinity and reduced Na^+^ accumulation within leaf tissues compared to uninoculated plants. Mutants defective in ACC deaminase, auxin catabolism, *N*-acyl-homoserine-lactone production, and flagellin synthesis showed a low relevance of these functions to salinity tolerance. Bacterial release of volatile organic compounds (mainly 2-undecanone, 7-hexanol, 3-methylbutanol and dimethyl disulphide) reproduced the effects of direct bacterial inoculation of roots, increasing plant growth rate and tolerance to salt stress conditions. Exposure of *A. thaliana* to different amounts of these molecules demonstrated their capability to affect growth, while exposure to a mixture of the first three compounds mimicked the effects of the bacterial strain on plant growth stimulation and salinity tolerance.	[73]
*Arachis hypogaea* L. (peanut)	*Klebsiella* sp., *Pseudomonas* sp., *Agrobacterium* sp., and *Ochrobactrum* sp. isolated from the halophyte *Arthrocnemum indicum*	Five diazotrophic salt-resistant strains of *Klebsiella*, *Pseudomonas*, *Agrobacterium,* and *Ochrobactrum* produced IAA and ACC deaminase, fixed N_2_, and solubilized phosphate. All of the isolates promoted peanut growth under non-stressful conditions and increased the N content in plants. In plants that were previously inoculated with these bacterial strains and then exposed to salt stress, accumulation of ROS-modulating enzymes and increased biomass was recorded compared to uninoculated ones.	[74]
*Brassica campestris* L. (canola)	*Brevibacterium epidermidis* RS15 and *Bacillus aryabhattai* RS341	120 mM NaCl reduced the rate of seed germination by 50%. Inoculation with *B. epidermidis* RS15 and *B. aryabhattai* RS34, both halotolerant and able to synthesize ACC deaminase, enhanced seed germination under salt stress and reduced the ACC content in seeds. Inoculation with both bacterial strains increased hydrolytic enzyme activities (amylase, invertase, and protease) and decreased ethylene levels compared to uninoculated seeds exposed to salt stress.	[75]
*Camelina sativa* (camelina or false flax)	*Pseudomonas putida* UW4, two root endophytes *Pseudomonas migulae* 8R6 and *Pseudomonas fluoresces* YsS6 (both ACC deaminase producing strains), and the *acdS* minus mutants 8R6M and YsS6M	Soil inoculation with wild-type strains increased shoot length without salt, and seed yield under moderate salinity. Transgenic plants that expressed the *acdS* gene, encoding the enzyme ACC deaminase, showed reduced inhibition of root lengthening and biomass development, and increased seed production, better seed quality, and higher levels of seed oil production under salt stress.	[76]
*Camelina sativa* (camelina or false flax)	*Pseudomonas migulae* 8R6	Both of the *C. sativa* plants treated with the ACC deaminase producing endophyte *P. migulae* 8R6 and transgenic plants expressing *acdS* demonstrated increased tolerance to salt. Inoculation with strain 8R6 positively impacted ethylene- and abscisic acid-dependent signalling. The expression of *acdS* in transgenic plants altered auxin, jasmonic acid, and brassinosteroid signalling and/biosynthesis. Expression of genes involved in carbohydrate metabolism were up-regulated, as was the expression of genes modulating the level of ROS released. The expression of the *acdS* gene also positively effected the expression of photosynthesis genes.	[77]
*Camelina sativa* (camelina or false flax)	*Pseudomonas migulae* 8R6	Treatment of *C. sativa*, grown under salt stress, with the endophyte *P. migulae* 8R6, able to synthesize ACC deaminase, induced a negative modulation of ethylene signaling as well as auxin and jasmonic acid biosynthesis and signaling, while genes involved in regulation of gibberellin signaling were positively affected. In plants cultivated with salt and inoculated with 8R6, a moderate expression of the *acdS* gene under the control of the *rolD* promoter occurred, which was highly efficient in lowering the expression of the genes involved in the synthesis of ethylene and its signaling.	[78]
*Capsicum annuum* (pepper)	*Brevibacterium iodinum* RS16, *Bacillus licheniformis* RS656, and *Zhihengliuela alba* RS111	*Brevibacterium iodinum* RS16, *Bacillus licheniformis* RS656, and *Zhihengliuela alba* RS111 were identified as both halotolerant and ACC deaminase producers. Single inoculation with the three bacterial strains in red pepper plants grown at three salinity levels induced lower ethylene production. Plant biomass and salt tolerance index (the ratio of the biomass of salt stressed to non-stressed plants) in inoculated plants was higher compared to non-inoculated plants.	[79]
*Capsicum annuum* L. (red pepper)	*Pseudomonas frederiksbergensis* OS261	Plants were inoculated with strain OS261 and grown with three levels of salt. Growth parameters (height and plant biomass) of plants were increased by the presence of the bacterial strain compared to uninoculated controls. The amount of ethylene synthesized by plants grown under salinity stress was high, but inoculation with strain OS261 reduced the release of this hormone. The level of antioxidant enzyme activity in leaves of inoculated plants grown in salinity was increased, while the H^+^ concentration was reduced.	[80]
*Capsicum annuum* L. cv. Bulmat (red pepper)	*Pseudomonas frederiksbergensis* OB139, *Pseudomonas vancouverensis* OB155	Plants were cultivated under four levels of salt concentration and inoculated or not with one or both strains. Salt stress inhibited plant growth through increased ethylene synthesis and the disruption of photosynthetic parameters compared to uninoculated plants. The combination of the two bacterial strains, both able to synthesize ACC deaminase, lowered ethylene levels in plants and increased catalase activity, leading to increased plant growth compared to a single bacterium or the uninoculated control.	[81]
*Cicer arietinum* L. (chickpea)	*Mesorhizobium ciceri* EE-7 (salt-sensitive) and *Mesorhizobium ciceri* G-55 (salt-tolerant)	Two isolates of *M. ciceri*, one that was salt sensitive and another that was salt tolerant, were transformed with an isolated *acdS* gene encoding ACC deaminase. Salt stress reduced the biomass of plants inoculated with the wild-type strains. The salt-tolerant bacterial strain induced a higher nodulation rate in chickpeas compared to the salt-sensitive strain. The shoot dry weight was increased in plants inoculated with the salt-sensitive transformant strain. In plants inoculated with the salt-sensitive transformant strain, nodulation was found to be comparable to that induced by the salt-tolerant strain.	[82]
*Coriandrum sativum* L. (coriander)	*Azospirillum brasiliense* and *Azotobacter chroococcum*	Inoculation of coriander seeds, exposed to four levels of salt stress, with a mixture of *A. brasiliense* and *A. chroococcum* enhanced chlorophyll content and increased grain yield and plant biomass compared to uninoculated plants. Combined inoculation and salt stress increased catalase and decreased the level of ascorbate peroxidase and guaiacol peroxidase compared to untreated plants. Inoculation with both PGPB lowered Na and increased the K concentration in coriander leaves compared to untreated plants. The presence of PGPB improved plant growth in both the absence and presence of salt stress conditions.	[83]
*Cucumis sativus* (cucumber)	*Pseudomonas fluorescens*, *Bacillus megaterium,* and *Variovorax paradoxus*	The ability to solubilize phosphates and synthesize ACC deaminase, siderophores, and IAA was assessed in the three PGPB strains grown at two salt concentrations (2 and 5% NaCl *w*/*v*). While *B. megaterium* was the least affected by high salinity, ACC deaminase activity as well as siderophore and IAA production in *P. fluorescens* remained unaffected under salt stress. On the contrary, *V. paradoxus* was not tolerant to salt, and its expression of plant beneficial traits was reduced by salinity. When inoculated onto cucumber plants grown at three different salinity levels, *P. fluorescens* was the most effective of the three strains at decreasing the inhibitory effects of salinity.	[84]
*Hordeum vulgare* L. (barley), *Trifolium repens* L. (clover), and *Pennisetum glaucum* L.R. Br. (pearl millet)	*Pseudomonas putida* UW3 and UW4	Barley, clover, and pearl millet plants grown in the presence of salt and inoculated with *P. putida* UW3 and UW4. *P. putida* UW4 increased barley biomass compared to uninoculated plants. Strain UW3 increased the biomass of the three crops. Shoot and root length and weight were increased in inoculated plants, suggesting a more efficient photosynthetic activity in the presence of the bacterial strains. Data from pulse amplitude modulation fluorometry showed that the reduction of plant photosynthetic activity induced by salt stress was recovered once the strains were applied.	[85]
*Medicago sativa* L. (alfalfa)	*Bacillus megaterium* NRCB001, *Bacillus subtilis* subsp. *subtilis* NRCB002, and *Bacillus subtilis* NRCB003	Thirteen bacterial strains were isolated from the rice rhizosphere and characterized for their plant beneficial traits. *B. megaterium* NRCB001, *B. subtilis* subsp. subtilis NRCB002, and *B. subtilis* NRCB003 synthesized auxin, siderophores, NH_3_, and ACC deaminase and solubilized phosphate and potassium. Strains NRCB001 and NRCB002 tolerated 1750 mM NaCl. The three strains were inoculated onto *M*. *sativa* grown under normal conditions and salinity stress. Strains NRCB002 and NRCB003 increased the dry weight of alfalfa compared with non-inoculated seedlings treated with 130 mM NaCl.	[86]
*Oryza sativa* L. (rice)	*Streptomyces* sp. GMKU 336 and its ACC deaminase-deficient mutant	Plants of Thai jasmine rice cultivar Khao Dok Mali 105 grown under salt stress were inoculated with the endophyte *Streptomyces* sp. GMKU 336 or with its mutant lacking ACC deaminase activity. Strain GMKU 336 increased plant growth and chlorophyll, proline, K^+^, Ca^+^, and water content. The amount of released ethylene was reduced, as was the content of ROS and Na^+^, and the Na^+^/K^+^ ratio, compared to uninoculated plants or to those inoculated with the mutant. Plants treated with the wild type showed down-regulation of genes involved in the ethylene synthesis pathway, *ACO1* and *EREBP1,* while *acdS* was up-regulated. Genes involved in osmotic balance, Na^+^ transport, calmodulin, and antioxidant enzymes were upregulated.	[87]
*Oryza sativa* (rice)	*Bacillus tequilensis* 10b (UPMRB9)	The effect of strains 10b UPMRB9′ on the growth of rice that was grown in the presence of salt was assessed. Strain 10b UPMRB9′ improved osmoprotectant properties such as proline, the soluble sugar concentration, and the levels of the antioxidant enzymes uperoxide dismutase, peroxidase, and catalase. Rice inoculated with strain UPMRB9 accumulated a greater amount of N and Ca in plant tissues, suggesting that this strain could behave as a bio-augmenter to improve biochemical and nutritional features in rice plants under salinity stress.	[88]
*Panax ginseng* (ginseng)	*Paenibacillus yonginensis* DCY84T	The impact of strain DCY84T, able to synthesize IAA and siderophore and solubilize phosphate, was assessed under short- and long-term salinity stress. Ginseng seedlings inoculated with the bacterial strain, following exposure to salt stress, were protected by the induction of plant defense-related systems (ion transport, ROS scavenging enzymes, proline content, total sugars, and ABA biosynthetic genes), as well as genes involved in root hair formation. The metabolome of the seedlings treated with DCY84T and exposed to salt stress overlapped with that of control plants.	[89]
*Pisum sativum* (pea)	*Bacillus marisflavi* (CHR JH 203) and *Bacillus cereus* (BST YS1_42)	Inoculation of pea plants with *B. marisflavi* CHR JH 203 and *B. cereus* BST YS1_42, both synthesizing a high amount of ACC deaminase, grown under salinity, improved plant biomass as well as the amount of plant carbohydrates, reducing sugars, proteins, chlorophylls, phenol, flavonoids, and antioxidant enzymes levels. In addition, plant ROS scavenging genes, defense genes, and cell rescue genes were all overexpressed in inoculated plants in the presence of 1% NaCl.	[90]
*Seidlitzia rosmarinus* Ehrenb. ex Boiss (perennial-green desert species of saltwort)	*Rothia terrae, Kocuria palustris, Pseudomonas baetica, Pseudomonas fluorescens Staphylococcus warneri, Staphylococcus epidermidis, Staphylococcus succinus, Paenibacillus amylolyticus, Brevibacterium frigoritolerans, Stenotrophomonas pavanii, Halomonas sulfidaeris, Planococcus salinarum, Planomicrobium koreense, Planococcus halocryophilus, Planomicrobium soli*	Culturable endophytic bacteria from the halophytic plant *Seidlitzia rosmarinus* Ehrenb. ex Boiss. were isolated and characterized to evaluate their plant beneficial traits under salt stress. Root endophytes belonged to genera *Rothia*, *Kocuria*, *Pseudomonas*, *Staphylococcus*, *Paenibacillus,* and *Brevibacterium*; shoot isolates belonged to *Staphylococcus*, *Rothia*, *Stenotrophomonas*, *Brevibacterium*, *Halomonas*, *Planococcus*, *Planomicrobium,* and *Pseudomonas* genera; *Staphylococcus*, *Rothia,* and *Brevibacterium* occurred in both roots and shoots. Synthesis of IAA and ACC deaminase was higher in bacteria from roots than from shoots. Finally, *S. pavanii* JST3 and *P. fluorescens* JST2 improved both shoot and root growth of Lepidium sativuum under salinity conditions.	[91]
*Solanum lycopersicum* (tomato)	*Pseudomonas azotoformans CHB* 1107	Strain CHB 1107 wild-type (producing ACC deaminase) lowered ethylene and proline levels in tomato plants exposed to high salt levels, increasing the dry weights of shoots and roots compared with uninoculated plants. Plants that were inoculated with a mutant that lacked ACC deaminase activity showed reduced K, Ca, and Mn uptake compared with plants inoculated with the wild-type strain. The wild-type strain CHB 1107 reduced the uptake of Na by tomato plants compared with the mutant strain under salt stress. Tomato plants inoculated with the wild-type strain yielded a higher K/Na ratio than those that were inoculated with the mutant.	[92]
*Sorghum vulgare* (sorghum)	*Pseudomonas migulae* SVB3R2, SVB3R3, SVB3R4, *Pseudomonas* sp. SVB3R5, *Pseudomonas brassicacearum* SVB6R1	Sorghum, tomato, and cucumber bacterial endophytes were characterized by 16S rRNA sequence determination and tested for plant beneficial traits. The activity of five endophytes was tested on plants grown with salinity stress. Strains SVB3R3 and SVB3R4 increased plant biomass, and strains SVB3R3 and SVB3R4 and SVB6R1 decreased the symptoms of plant salinity stress. Only strain SVB6R1 could produce ACC deaminase.	[93]
*Triticum aestivum* (wheat)	*Bacillus pumilus* SU3, *Bacillus aquimaris* SU8, *Bacillus pumilus* SU10, *Bacillus arsinicus* SU13, *Arthrobacter* sp. SU18, *Bacillus cereus* SU24, *Pseudomonas mendocina* SU40, *Bacillus aquimaris* SU44, and *Bacillus subtilis* SU47	Salt-tolerant (ST) PGPB positively influenced the growth and yield of wheat in saline soil. All nine tested strains improved plant growth in saline soil under greenhouse conditions, with strain DU18 being the most efficient. Under field conditions, strains SU44 and SU8 were the best in increasing plant biomass. Plant inoculation with strain SU8 led to higher proline content and total soluble sugar accumulation in wheat, while strain SU44 resulted in a higher accumulation of reducing sugars. The amounts of N, K, and P in wheat leaves increased significantly after inoculation with all the strains; *B. subtilis* SU47 lowered the sodium (Na) content in wheat leaves.	[94]
*Triticum aestivum* (wheat)	*Serratia marcescens* CDP-13	*Serratia marcescens* CDP-13 is halotolerant, produces ACC deaminase, solubilizes phosphate, synthesizes siderophores and IAA, and fixes N_2_. Wheat inoculation with strain CDP-13 increased plant biomass under salinity stress, reducing inhibition of plant growth caused by salt and lowering the amount of osmoprotectants (such as proline, malondialdehyde, soluble sugar), protein, and IAA content in plants.	[95]
* Triticum **aestivum* L. (wheat).	* Bacillus **megaterium* PN89	*B. megaterium* PN89, able to synthesize IAA, induced increased germination rate and root and shoot length in wheat plants exposed to salt stress, compared to non-inoculated controls.	[96]
*Triticum aestivum* (wheat)	*Brevibacterium frigoritolerans, Bacillus velezensis,* and *Bacillus thuringiensis*	The expression of plant-beneficial traits of *Br. frigoritolerans* alone or in combination with *B. velezensis* and *B. thuringiensis* under six salinity levels was characterized, and the effects on wheat of these strains alone or in combination under salt stress were assessed. *B*. *frigoritolerans* was the most effective, both for physiological trait expression and wheat plant growth promotion. Under salinity stress, the mixed inoculation of the three bacterial strains was more efficient than any single inoculation.	[97]
*Triticum durum* (wheat)	Fourteen strains of the genera *Streptomyces* and *Nocardiopsis*	Fourteen Actinomycetes strains were tested for expression of plant-beneficial activities under salinity conditions. The isolates could solubilize inorganic phosphate and synthesize IAA, HCN, and ammonia when grown in the presence of different salt concentrations. The majority of the strains produced ACC deaminase. Plant inoculation with these strains improved biomass and yielded an increased amount of chlorophyll and proline compared to uninoculated plants, both with and without salt.	[98]
*Vigna radiata* L. (mung bean)	*Enterobacter cloacae* KBPD	Strain KBPD is an ACC deaminase producer, able to solubilize phosphates and synthesize IAA, siderophore, ammonia, hydrogen cyanide, and exopolysaccharide. *V. radiata* plants exposed to salinity and inoculated with this bacterial strain showed increased shoot length, root length, and fresh and dry weights. Inoculation with strain KBPD also reduced proline content in plants grown with salt stress.	[99]
*Vigna radiata* (L.) R. Wilczekspring (mung bean)	*Rhizobium* sp. LSMR-32 and *Enterococcus mundtii* LSMRS-3	Under salt stress conditions, separate inoculation with the two bacterial strains induced increased seed germination, grain yield, plant height, biomass, chlorophyll content, and nutrient uptake compared to uninoculated plants. Inoculation with both strains increased both symbiotic parameters (nodulation rate, nodule biomass, and leghaemoglobin amount) and soil phosphatase and dehydrogenase levels. The microbial consortium enhanced the level of proline and anti-oxidative enzymes.	[100]
*Zea mays* (maize)	*Azospirillum lipoferum* or *Azotobacter chroococcum*	Maize (corn) plants that were exposed to salt stress had reduced growth parameters, pigments, soluble proteins, K^+^, and a K^+^/Na^+^ ratio. Salinity led to increased levels of soluble sugars, proline, Na^+^, malondialdehyde, and peroxidase and catalase activity, while the activity of plant ascorbate peroxidase remained unaffected. Plants inoculated with *A. lipoferum* or *A. chroococcum* increased growth parameters, pigments, K^+^, osmolytes, K^+^/Na^+^ ratio, and the antioxidative enzymes in salt-affected maize. Both bacterial strains also lowered malondialdehyde and Na^+^ in maize plants.	[101]

The majority of the papers reported in Table 1 and Table 2 are focused on experiments performed under controlled laboratory or greenhouse conditions. Unfortunately, despite their effectiveness under these conditions, the field application of PGPB covers only a small portion of agriculture at a global level [102]. Although PGPB are more acceptable for deliberate use in the field than transgenic plants in many countries, their exploitation is well below their potentiality, and this is mainly related to the lack of consistency and repeatability of PGPB performance in open field conditions. The survival of PGPB once inoculated onto the plants is one of the main problems. This is related to the compatibility of the selected PGPB with the considered plant species and the soil characteristics. Once introduced in the environment, the PGPB establishes relationships with the resident microbiota, and the results of this interaction may range from positive to negative [103]. Finally, the inoculant formulation has a specific role in maintaining the strain survival, especially after inoculation in open field conditions [104] Regarding stressed soil such as those affected by an excess of salt or water deficiency, it is necessary to consider the adaptability of the selected PGPB to these harsh conditions. When considering a stressed environment, it is always important to keep in mind that the expression of plant-beneficial bacterial physiological traits can be modified by the stress itself [105,106]. Following the analysis of the literature for this manuscript, it is clear that there is a general lack of data on the performance of PGPB in increasing plant growth and yield under salinity and drought stress conditions under open field conditions.

## 3. Groups of Microorganisms to Lower Plant Drought and Salt Stress

Since bacteria can express different plant-beneficial activities, inoculation of plants with a combination of two or more non-antagonistic microorganisms, belonging to the same or different species, or including both prokaryotic and eukaryotic organisms, can often lead to additive or synergistic effects in the promotion of plant growth. In fact, the lack of one plant beneficial activity in one strain can sometimes be overcome if another microorganism expresses this specific physiological trait [29,107,108,109]. Moreover, the combination of diverse plant growth-promoting microorganisms (PGPM) offers a wide range of positive activity to a plant, including the enhancement of plant growth, reduced susceptibility to soil-borne diseases, increased yield, and improved seed and fruit nutritional value. The commercialization of groups of PGPM as biofertilizers and biocontrol agents represents the bottleneck of the full process [110,111]. However, as demonstrated by the large number of scientific papers recently published on this topic, the use of microbial consortia is becoming a reliable tool to support plant growth in stressed environments, especially those characterized by an excess of salt [112,113,114,115,116] or a deficiency of water [63,117,118,119,120,121].

### 3.1. Bacterial Consortia

Bacterial consortia may be classified as either simple or complex. In simple consortia, all of the bacterial strains are inoculated in a bioreactor at the same time, while in a complex consortium, the bacterial growth occurs in separate bioreactors [29,122]. Once the strategy of employing a simple consortium has been decided upon, it should be taken into account that the metabolism of each inoculated bacterial strain affects the growth and physiology of the other co-existing strains, leading to a plethora of possible interactions, ranging from neutral, to positive (cooperation, synergism, commensalism), to negative (amensalism, competition, antagonism). Although the occurrence of one or the other type of interactions is usually finely monitored and regulated during the scale-up of the fermentation process, the issue of the strain’s compatibility plays a pivotal role in the final result [123]. For, example, in *Pseudomonas koreensis* S2CB45 (which was isolated from the surface sterilized spore of a mycorrhizal fungi) and *Microbacterium hydrothermale* IC37-36 (which was isolated from a rice spermosphere), the expression of plant-beneficial traits such as ACC deaminase, IAA, and cytokinin was higher when these PGPB were co-cultured rather than grown separately. Consequently, when this consortium was used to inoculate red pepper plants grown at two salt stress levels, these bacterial strains reduced ethylene emission by 20% compared to uninoculated plants, lowered ROS accumulation, and increased activity of antioxidant enzymes [112].

Especially for those consortia intended for use as plant growth promoters in saline or arid environments, the sampling site from which the bacterial strains are isolated is key. Since autochtonous plants growing in arid environments have co-evolved with their associated microbiota, it is a common opinion that selecting bacterial strains from these specific environments may help to obtain well-adapted bacterial strains with high tolerance to stress as well as long survival and persistence in open field conditions [124,125].

In a recent study, Mansour et al. [118] examined the effects of either the single or double inoculation of five faba bean cultivars with strains *R. leguminosarum* bv. *viciae* USDA 2435 and *P. putida* RA MTCC5279 on the growth and health of these plants treated to (i) optimal watering, (ii) moderate watering, or (iii) severe drought conditions. While the water deficiency reduced the synthesis of photosynthetic pigments and enhanced the production of antioxidant enzymes and osmoprotectants to a different extent according to the drought sensitivity of the cultivar, plant inoculation with the bacterial strains led to increased seed yield and crop water productivity.

Similarly, among 89 strains originally from the rhizosphere or endosphere of two Algerian autochthonous halophytic plants (*Suaeda mollis* and *Salsola tetrandra*), three bacterial strains, i.e., *Bacillus atropheus* (BR5, OR15, and RB13), were selected based on plant growth-promoting traits (IAA, ACC deaminase, nitrogen fixation, and phosphate solubilization), antifungal activity, and tolerance to environmental conditions (pH, PEG, and NaCl). The bacterial strains were used to inoculate *A. thaliana* and durum wheat alone or in combination under salt stress conditions. Overall, the data obtained indicate that bacterial inoculation of both plant species increased plant growth under normal conditions. When wheat plants were cultivated in soils characterized by three salinity levels, inoculation with the consortium was the most efficient strategy for increasing growth compared to control plants. Finally, inoculation with the consortium increased chlorophyll and carotenoid contents while lowering proline content, lipid peroxidation, and the activities of antioxidant enzymes in treated plants, suggesting that the bacterial treatments limit the damages induced by salt stress [116].

### 3.2. Bacteria Plus Fungi (Including AMF)

Consistent with the fact that plant microbiota includes a wide variety of different microorganisms, the effects of consortia involving bacterial strains associated with arbuscular mycorrhizal fungi (AMF) and other non-obligate symbiotic fungi on plant growth under arid conditions is a powerful tool, where each microorganism potentiates the bioprotection against the effects of various abiotic stresses on the plant [29]. AMF are common plant-beneficial fungi that typically form a symbiotic relationship with the roots of ~80–90% of all land plants, colonizing plant roots intracellularly. AMF enter the root cortical cell walls, and once inside these cells, form obligate branched intracellular fungal structures (arbuscules), where an exchange of nutrients between the plant and the beneficial fungus takes place. In this exchange, the plant provides the fungus with both fixed carbon and nitrogen, and in exchange, the fungus provides the plant with an effectively increased root system and a significantly increased ability to take up water and nutrients from the soil.

The work of several researchers has highlighted the occurrence of additive and synergistic effects between AMF and PGPB [29], with PGPB behaving as mycorrhizal helper bacteria and thus favoring the establishment of the mycorrhizal symbiosis. Even more complex is the interrelationship involving legumes, rhizobia, and AMF, where the AMF can favor nodule formation, while rhizobia may reduce mycorrhizal hyphal development [126]. On this latest point, contrasting results have been obtained by Igiehon and Babalola [121], who assessed the effects of a combination of rhizobia species and AMF on the growth of soybean under drought stress. Based on their results, co-inoculation of soybean plants with the bacteria *Rhizobium* sp. R1 and *R. cellulosilyticum* R3, together with a consortium of AMF (consisting of *Paraglomus occulum*, *Gigaspora gigantea*, *Funneliformis mosseae*, *Claroideoglomus etunicatum* and *Rhizophagus clarus),* improved the growth and yield of soybean plants that were exposed to 40% water stress, leading to enhanced leaf relative water content and proline concentration. Moreover, mycorrhizal plants that were inoculated with rhizobia displayed the greatest number of fungal spores and mycorrhizal colonization intensity of all of the water regimes, revealing that the rhizobia and AMF consortium can positively interact and represent a valuable tool in sustainable agriculture.

A large number of factors can affect the tripartite relationship between the host plant, mycorrhizal fungi, and PGPB strains, ranging from the host plant and AMF species involved to the chemical and physical parameters of the growth substrate(s) [127,128]. For example, Veselaj et al. [129] reported that in legumes the relationships occurring among AMF, PGPB, and rhizobia are dependent both on the species of microorganisms involved and the level of salinity. In this study, inoculated pea plants were grown at two different levels of salt stress, with two AMF (*Rhizophagus irregularis* and *C. claroideum*) and/or two PGPB, one of them being a rhizobial strain (*Rhizobium leguminosarum*) and *Burkholderia* sp. When grown under optimal conditions, the plant growth parameters were improved by inoculation with *R. irregularis* and bacteria *(R. leguminosarum* and *Burkholderia* sp.) or by the combination of the AMF (*R. irregularis* + *C*. *claroideum*) with each type of bacteria. While the plant growth was reduced by the salt stress, inoculation with *R. irregularis* led to better vegetative development and higher productivity than with *C. claroideum*. Moreover, under salt stress, pea plants treated with *Burkolderia* sp. increased vegetative growth more efficiently than *R. leguminosarum*. In this experiment, the best results, in terms of plant growth and productivity, were observed with plants inoculated with both AMF the rhizobia strain.

In addition to AMF, mixed bacterial/fungal consortia can include other microfungi, such as isolates of *Trichoderma* species and the root endophyte *Serendipita indica* (previously classified as *Piriformospora indica)*. While *Trichoderma* spp., and especially *Trichoderma harzianum*, is mainly known for its mycoparasistic behaviour mediated by chitin degradation [130], *S. indica* is a root endophyte mimicking AMF, able to grow in pure culture and improve nutrient uptake, increasing disease resistance and stress tolerance to environmental conditions [131]. In a recent study [132], lettuce and tomato plants were cultivated in the presence of different salinity levels and inoculated with two commercial biostimulants, one containing *Bacillus amyloliquefaciens, B. brevis, B. circulans, B. coagulans, B. firmus, B. halodenitrificans, B. laterosporus, B. licheniformis, B. megaterium, B. mycoides, B. pasteurii, B. subtilis,* and *Paenibacillus polymyxa,* and the other *Glomus* spp., *Agrobacterium radiobacter*, *Bacillus* subtilis, *Streptomyces* spp., and *Thricoderma* spp. While uninoculated plants showed symptoms related to salinity, plants inoculated with the two formulations showed increased biomass, leaf number, and leaf area and were less sensitive to salinity stress. Comparing the efficacy of the two commercial formulations, the inoculum with the consortium containing AMF was more effective than the one based exclusively on *Bacillus* species.

Notwithstanding the good results obtained inoculating plants with *S. indica,* both under optimal and drought/salt stressed conditions [133,134,135,136,137], combinations of *S. indica* and PGPB have not always yielded positive results. Thus, the possibility of using this fungus in consortia with PGPB has been shown to be effective only in studies aimed at increasing plant disease resistance [138].

## 4. Summary and Conclusions

Approximately 20 years ago, scientists began observing and reporting that PGPB could help plants to overcome some of the effects of drought and salt stress. Since that time, there have been a large number of literature reports confirming the ability of PGPB, applied by itself or as part of a consortium, to ameliorate some of the negative effects of the related abiotic stresses of drought and salt stress on plant growth and development. On the one hand, PGPB can directly promote plant growth, thereby helping plants to defend themselves against growth-inhibiting drought and salt stress. On the other hand, PGPB can augment a plant’s defenses against the often deleterious effects of drought and/or salt stress using a number of different mechanisms. These mechanisms include (but are not limited to) the synthesis of the enzyme ACC deaminase, which can decrease plant stress ethylene levels, the synthesis of IAA and cytokinin, which can modulate a plant’s response to stress and promote its growth, and the synthesis of various osmoprotectant molecules, such as proline and trehalose, which can structure plant intracellular water, thereby protecting the cell’s structural integrity.

To date, the majority of the reported experiments using PGPB to ameliorate drought and salt stress include just a single well-characterized PGPB strain. However, scientists have demonstrated that the addition of several bacterial strains or a mixture of bacterial and fungal strains may also be highly effective. However, additional work is required to determine whether some of these consortia will be sufficiently stable and therefore effective throughout a growing season and in the long term. In addition, just how effective these approaches will be in the field (as opposed to in growth chamber and greenhouse experiments) in the presence of severe drought or salt stress remains to be determined. Certainly, the increased use of naturally occurring PGPB in agriculture as a means of addressing various abiotic stresses is an attractive approach that (based on the results that have been published) shows quite a lot of promise. Nevertheless, there are still many scientific and regulatory hurdles to overcome before this technology is widely accepted and used on a large scale.

## Figures and Tables

**Figure 1 biology-11-00437-f001:**
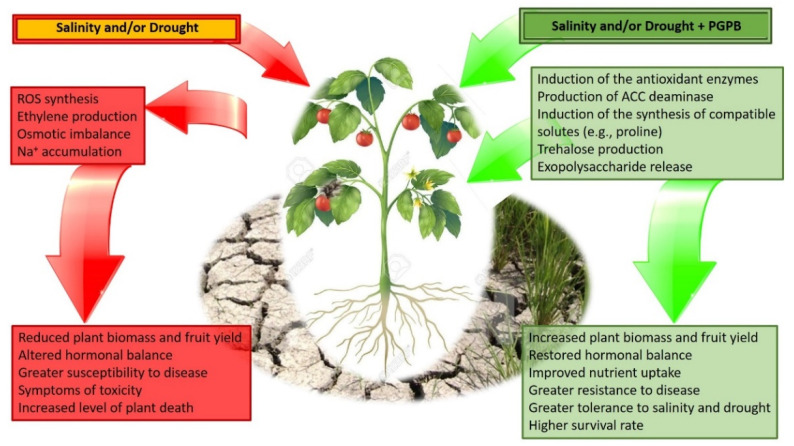
Schematic overview of the consequences of salt and drought stress on plants and the functioning of PGPBs supporting plant growth in these harsh environmental conditions.

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
