# Peer review of "Recent Advances in Bacterial Amelioration of Plant Drought and Salt Stress"

_biology, 2022, doi:10.3390/biology11030437_

Round 1

Reviewer 1 Report

The manuscript by Gamalero and Glick reviews the role of plant growth-promoting bacteria (PGPB) in ameliorating the abiotic plant stress caused by drought and salinity. As stated by the authors, the responses to these two stresses share some common mechanisms, and relevant research efforts have been done in the latest years towards a better understanding of which molecular mechanisms account for the beneficial effects of PGPB on these stress conditions.

The manuscript is well organized and covers the latest studies in this area. The only suggestion I would make is the addition of a paragraph or section dedicated to field trials using PGPB to alleviate plant drought and salt stress.

I find the information included in Tables 1 and 2 very useful, although I have the following minor comments:

- organize the plant species column using alphabetic order, so that multiple studies using the same plant would be in consecutive rows

-always include the plant common name (even if repeated)

-use the species full name in the bacteria column (genera abbreviation may then be used in the comments column)

-in the row corresponding to reference 59: “(walnut)” and “AM fungi (Funnelliformis mosseae, Claroideoglomus etunicatum), Azotobacter chroococcum and Azospirillium lipofrum

-in the row corresponding to reference 62: please state the bacterial species

-in the row corresponding to reference 80: “three PGPB strains grown at two salt”

- in the row corresponding to reference 92: please move the species names from the comments column to the bacteria column; and add the information that isolates S. pavanii JST3 and P. fluorescens JST12 significantly improved root and shoot growth of Lepidium sativum under salinity conditions.

Other minor comments

-lines 25 – “structure the water content within”

-line 106 – “numerous works”

-line 131 – “as a nutrient source”

-line 145 – “PGPB act as”

-line 146 – “and prevent or thwart”

-line 244 – “archaea”

-line 246 – “Pseudomonas putida

-line 248 – “hypothesized”

-line 299 – “enhancement”

-lines 353 – “Bacteria plus fungi (including AMF)

-line 357 – “potentiates the bioprotection against the effects”

-lines 373-374  – “of AMF (consisting of”

-line 400 – “behaviour”

-line 412 – “containing AMF was”

Author Response

Reviewer 1

The manuscript by Gamalero and Glick reviews the role of plant growth-promoting bacteria (PGPB) in ameliorating the abiotic plant stress caused by drought and salinity. As stated by the authors, the responses to these two stresses share some common mechanisms, and relevant research efforts have been done in the latest years towards a better understanding of which molecular mechanisms account for the beneficial effects of PGPB on these stress conditions.

The manuscript is well organized and covers the latest studies in this area. The only suggestion I would make is the addition of a paragraph or section dedicated to field trials using PGPB to alleviate plant drought and salt stress.

A: We thank the reviewer, and we agree with this suggestion. Therefore, we have added a paragraph regarding inconsistent performance of PGPB in field condition after the two tables (PAGE 32 LINE 346-367 of the version with tracking changes)

The majority of the papers reported in Tables 1 and 2 are focused on experiments performed under controlled laboratory or greenhouse conditions. Unfortunately, despite their effectiveness under these conditions, the field application of PGPB covers only a small portion of agriculture at a global level [102]. Although PGPB are more acceptable for deliberate use in the field than transgenic plants in many countries, their exploitation is well below their potentiality, and this is mainly related to the lack of consistency and repeatability of PGPB performance in open field conditions. The survival of PGPB once inoculated onto the plants is one of the main problems. This is related to the compatibility of the selected PGPB with the considered plant species and the soil characteristics. Once introduced in the environment the PGPB establishes relationships with the resident microbiota and the results of this interaction may range from positive to negative [103]. Finally, the inoculant formulation has a specific role in maintaining the strain survival especially after inoculation in open field conditions [104] Regarding stressed soil such as those affected by an excess of salt or water deficiency, it is necessary to consider the adaptability of the selected PGPB to these harsh conditions. When considering a stressed environment, it’s always important to keep in mind that the expression of plant-beneficial bacterial physiological traits can be modified by the stress itself [105,106]. Following the analysis of the literature for this manuscript, it is clear that there is a general lack of data on the performance of PGPB in increasing plant growth and yield under salinity and drought stress conditions under open field conditions.

I find the information included in Tables 1 and 2 very useful, although I have the following minor comments:

- organize the plant species column using alphabetic order, so that multiple studies using the same plant would be in consecutive rows

A: This has been done, as suggested by the referee

-always include the plant common name (even if repeated)

A: This has been done, as suggested by the referee

-use the species full name in the bacteria column (genera abbreviation may then be used in the comments column)

A: This has been done, as suggested by the referee

-in the row corresponding to reference 59: “(walnut)” and “AM fungi (Funnelliformis mosseaeClaroideoglomus etunicatum), Azotobacter chroococcum and Azospirillium lipofrum

A: This has been done, as suggested by the referee

-in the row corresponding to reference 62: please state the bacterial species

A: We can’t add this information since the Authors identified only the most performant strain out of the 12 selected

-in the row corresponding to reference 80: “three PGPB strains grown at two salt”

A: This has been done, as suggested by the referee

- in the row corresponding to reference 92: please move the species names from the comments column to the bacteria column; and add the information that isolates S. pavanii JST3 and P. fluorescens JST12 significantly improved root and shoot growth of Lepidium sativum under salinity conditions.

A: This has been done, as suggested by the referee

Other minor comments

-lines 25 – “structure the water content within”

A: This has been done, as suggested by the referee

-line 106 – “numerous works”

A: This has been done, as suggested by the referee

-line 131 – “as a nutrient source”

A: This has been done, as suggested by the referee

-line 145 – “PGPB act as”

A: This has been done, as suggested by the referee

-line 146 – “and prevent or thwart”

A: This has been done, as suggested by the referee

-line 244 – “archaea”

A: This has been done, as suggested by the referee

-line 246 – “Pseudomonas putida

A: This has been done, as suggested by the referee

-line 248 – “hypothesized”

A: This has been done, as suggested by the referee

-line 299 – “enhancement”

A: This has been done, as suggested by the referee

-lines 353 – “Bacteria plus fungi (including AMF)

A: This has been done, as suggested by the referee

-line 357 – “potentiates the bioprotection against the effects”

A: This has been done, as suggested by the referee

-lines 373-374  – “of AMF (consisting of”

A: This has been done, as suggested by the referee

-line 400 – “behaviour”

A: This has been done, as suggested by the referee

-line 412 – “containing AMF was”

A: This has been done, as suggested by the referee

Reviewer 2 Report

The paper is well written and interesting to read. I therefore recommend acceptance and its publication by the journal.

Author Response

Reviewer 2

The paper is well written and interesting to read. I therefore recommend acceptance and its publication by the journal.

A: We thank the reviewer 2 for their positive opinion of our manuscript.

Reviewer 3 Report

Already so much of work has been done on the amelioration of plant drought and salt stress by bacteria. especially the PGPR. The authors themselves clearly stated in summary that 20 years back only the work has been started on the similar topic. Lack of novelty, authors could have opted to explain more advanced techniques to explain the abiotic stress tolerance in plants by bacteria.

Supplementary comments:

1. As I stated earlier also, I could not find any advanced mechanisms explaining the amelioration of drought and salt stress in plants by bacteria. The topics covered in the present paper were already explained by many researchers and thousands of papers are already available online. 
2. I found Table 1 interesting.and explained very well about the inoculation and mechanisms.
3. In my opinion authors may have explained the success rate of these bacteria in field conditions, which is now a priority. 
4. They may even have considered the formulation technologies and smart delivery mechanisms of products developed by plant growth promoting bacteria.

Author Response

Reviewer 3

Already so much of work has been done on the amelioration of plant drought and salt stress by bacteria. especially the PGPR. The authors themselves clearly stated in summary that 20 years back only the work has been started on the similar topic. Lack of novelty, authors could have opted to explain more advanced techniques to explain the abiotic stress tolerance in plants by bacteria.

A: The aim of this review was to synthesize and analyze previous recently published research. We have attempted to put that work into an appropriate context for the reader.

Supplementary comments:

  1. As I stated earlier also, I could not find any advanced mechanisms explaining the amelioration of drought and salt stress in plants by bacteria. The topics covered in the present paper were already explained by many researchers and thousands of papers are already available online. 

A: In our opinion, the advanced mechanisms are not bacterial in nature. In that regard, the literature is filled with various plant genes/proteins that affect salt and/or drought stress. We believe that we have well covered all of the recent mechanisms used by bacteria (trehalose being the most unique) both in the text and in the tables. Of course, future developments may come from partnering PGPB with appropriate activities together with transgenic plants.

  1. I found Table 1 interesting. and explained very well about the inoculation and mechanisms.

A: We thank the reviewer for this comment.

  1. In my opinion authors may have explained the success rate of these bacteria in field conditions, which is now a priority. 

A: We agree with the reviewer on this point. In fact, we have added a paragraph regarding the inconsistent performance of PGPB in field conditions. This paragraph is inserted into the text after the two tables (PAGE 32 LINE 346-367 of the version with tracking changes).

The majority of the papers reported in Tables 1 and 2 are focused on experiments performed under controlled laboratory or greenhouse conditions. Unfortunately, despite their effectiveness under these conditions, the field application of PGPB covers only a small portion of agriculture at a global level [102]. Although PGPB are more acceptable for deliberate use in the field than transgenic plants in many countries, their exploitation is well below their potentiality, and this is mainly related to the lack of consistency and repeatability of PGPB performance in open field conditions. The survival of PGPB once inoculated onto the plants is one of the main problems. This is related to the compatibility of the selected PGPB with the considered plant species and the soil characteristics. Once introduced in the environment the PGPB establishes relationships with the resident microbiota and the results of this interaction may range from positive to negative [103]. Finally, the inoculant formulation has a specific role in maintaining the strain survival especially after inoculation in open field conditions [104] Regarding stressed soil such as those affected by an excess of salt or water deficiency, it is necessary to consider the adaptability of the selected PGPB to these harsh conditions. When considering a stressed environment, it’s always important to keep in mind that the expression of plant-beneficial bacterial physiological traits can be modified by the stress itself [105,106]. Following the analysis of the literature for this manuscript, it is clear that there is a general lack of data on the performance of PGPB in increasing plant growth and yield under salinity and drought stress conditions under open field conditions.

  1. They may even have considered the formulation technologies and smart delivery mechanisms of products developed by plant growth promoting bacteria.

A: Regarding this specific point we think that formulation procedures, bacterial encapsulation, as well as smart delivery systems, are topics deserving of an entire review article by itself and is way beyond the scope of this review.

Round 2

Reviewer 3 Report

NA